# FIB and Wedge Polishing Sample Preparation for TEM Analysis of Sol-Gel Derived Perovskite Thin Films

Jorge Sanz-Mateo [1,2,]*, Marco Deluca [1], Bernhard Sartory [1], Federica Benes [1] and Daniel Kiener [2]

1 Materials Center Leoben Forschung GmbH, 8700 Leoben, Austria; marco.deluca@mcl.at (M.D.); bernhard.sartory@mcl.at (B.S.); federica.benes@mcl.at (F.B.)

2 Department of Materials Science, Montanuniversität Leoben, 8700 Leoben, Austria; daniel.kiener@unileoben.ac.at

* Correspondence: jorge.sanzmateo@mcl.at; Tel.: +43-681-8469-2310

**Abstract:** In ceramic thin films, choosing an appropriate sample preparation method for transmission electron microscopy (TEM) analyses is of paramount importance to avoid preparation-induced damage and retain nanoscale features that require investigation. Here we compare two methods of TEM thin film sample preparation, namely conventional wedge polishing and focused ion beam (FIB) based lift out preparation applied to ferroelectric barium titanate (BaTiO$_3$, BT) thin films made by chemical solution deposition (CSD). The aim of the work is to determine the pros and cons of each method considering not only the quality of the TEM specimen, but also aspects such as availability, ease of use, and affordability. Besides some limitations on the selection of visualized area due to thickness constraints on the FIB-made sample, both methods offer the capability to prepare samples with very comparable quality, as indicated by achieving the same thickness, a largely agreeing microstructure, no secondary phases on the diffraction pattern, and good atomic resolution. This last observation is especially important in the current context of material science, where more nanoscale phenomena are becoming the subject of study. The wedge polishing method, however, is deemed more affordable in terms of instrumentation, as it only requires a tripod polisher, a polishing wheel, and a precision ion polishing system, whereas the lift out method requires a scanning electron microscope (SEM) equipped with an FIB system. We believe that this work serves groups working on ferroelectric thin films in preparing TEM samples in a more effective and uncomplicated manner, facilitating progress in understanding this fascinating class of materials.

**Keywords:** transmission electron microscopy (TEM); wedge polishing; focused ion beam (FIB); thin films





## 1. Introduction

Among the different methods of materials characterization, transmission electron microscopy (TEM) is one of the most versatile. It can determine the structure of a material, its defects and chemical composition at different scales, from the micrometer [1] to the nanoscale [2]. Its results can also complement other methods of characterization, such as X-ray diffraction (XRD) [3] or Raman [2]. The analysis of materials at atomic resolution is becoming more relevant with the development of miniaturized systems in multiple domains such as electronics [4], chemistry [5] or metallurgy [6]. Most of these systems rely on the use of nano-engineered materials such as thin films [7] or more complex patterned structures [8]. Although the TEM community developed and still pushes further on a multitude of instruments [9,10] well suited to studying these systems, the requirements of thinner and cleaner samples made by an adequate preparation is still a key prerequisite to acquiring meaningful images and spectroscopic data for further analysis. The more the scale of the observed phenomena is reduced, the better the quality of the sample needs to be [11]. Despite the existence of multiple preparation techniques for different materials [12], TEM sample preparation is often a major experimental challenge. The difficulty comes,

most of the time, from choosing the best method that provides a very thin, non-altered slice of the material, while at the same time being reproducible, reliable and affordable.

Focusing on the field of thin film ceramics, this is not so different. Methods such as grinding and polishing [13–15] or dimple grinding [16,17] followed by energy ion milling or focused ion beam (FIB) [18,19] are well established when working with such materials, being able to achieve thin samples suitable for atomic resolution investigations. Because the volume of interest is reduced when studying thin films (in comparisons with their bulk counterparts), more accuracy and skill are needed to prepare the sample, even more so for cross-sectional analysis compared to plane view investigations. Although all the methods are feasible, their advantages or disadvantages must be considered when choosing one. For example, FIB has the advantage of being accurate, as the formation of the sample is being monitored in situ thanks to the scanning electron microscope (SEM) attached to the instrument. However, then the sample can be contaminated by the Ga employed during the thinning [20]; in such cases, a thermal annealing might be able to remove this undesired material [21]. To this we must add that an SEM system with an FIB attached is relatively expensive and requires regular maintenance and adjustment by certified personal. On the other hand, mechanical methods such as tripod polishing might appear more accessible, but problems can arise from the mechanical loading during preparation as well as during the final ion milling step, since the $Ar^+$ ions can modify the original material, whether by changing the phase [22], rendering it amorphous or redepositing some material [12].

Among the different methods for growing ceramic thin films, chemical solution deposition (CSD) [23] has the benefit of being a low cost process offering high scalability, which is potentially interesting for researchers and industries working in electronic applications [24]. These films can also be prepared for TEM analysis by different methods such as dimpling [16], cross-sectional wedge polishing [25] or FIB [26]. These films tend to grow forming multiple grains in a columnar manner, their width being around 100 nm, which affects their electric properties [27]. It is possible to create epitaxial thin films, with methods such as pulsed laser deposition (PLD) [28], but it is more demanding to scale them for mass production, as the equipment required is more complex and expensive than for the CSD case [29].

In this work, we compare two of the most popular sample preparation methods for the challenging question of cross-sectional thin film analysis: wedge polishing followed by a short session of ion-milling [30] and conventional FIB lift out preparation [31], respectively. We apply these two procedures to $BaTiO_3$ thin film samples made by the CSD method. The two procedures are compared in terms of the extent and homogeneity of the area accessible for analysis as well as the structural quality of the sample when examined in detail in TEM. The specimen is studied both at the microscale, checking the grain morphology and formation, and also at its atomic scale with high resolution TEM (HRTEM). As mentioned before, the ability to analyze nanoscale films at atomic resolution images is becoming more important with the demand of miniaturized devices and the study of nanoscale phenomena. The resources required to actually prepare such samples are also compared and commented on.

## 2. Materials and Methods

### 2.1. Thin Film Deposition

The commercially available substrate used consisted of Si with a native $SiO_2$ layer (500 nm), followed by $TiO_2$ (30 nm) and finally Pt (100 nm) at the top, oriented along the [111] axis (SINTEF, Trondheim, Norway). The $BaTiO_3$ solution [23] was prepared by dissolving barium acetate (Merck/Sigma-Aldrich, 99%, Darmstadt, Germany) and titanium isopropoxide (Merck/Sigma-Aldrich, 99%) respectively in acetic acid (Carl Roth, 100%, Karlsruhe, Germany) and 2-methoxyethanol (Carl Roth, 99%) in stoichiometric ratios. The precursor solutions were stirred until homogeneous in a dry glovebox. After mixing them, the volume was adjusted to obtain a solution with a concentration of 0.3 M, which was stored in the fridge. The prepared $BaTiO_3$ sol-gel precursor solution was diluted down to

0.07 M and deposited on the substrate by spin coating and was annealed at 800 °C for 5 min in a rapid thermal annealer (MILA-5050, ULVAC, Munich, Germany) to obtain a crystalline layer. This procedure was repeated 18 times. After the last deposition, the system was annealed one last time at 800 °C for an hour. Gold dot electrodes were deposited later via e-beam evaporation using a KORVUS deposition system (Maidenhead, UK) and a shadow mask. A picture of the final sample can be seen at the top of the Figure 1, the electrodes being the golden circles evident on the surface.

### 2.2. Cross Section Wedge Polishing

For this method, the specimen was cut and glued face-to-face forming a sandwich with heat curing resin (Allied High Tech. Inc, Compton, CA, USA), and further cut as a small square of 1–1.3 mm [32]. The next steps were performed on a multiprep machine (Allied High Tech. Inc, Compton, CA, USA). A thinning pad, previously planarized relative to the polishing wheel, was used to fix the sample. To glue the sample to the pad, heated wax was used. First, the wedge limit side was polished with diamond foil pads, each time decreasing its grain size (from 30 to 9, then 3 and lastly 1 µm). The same was subsequently performed on an adjacent side. Once finished, the sample was flipped on the thinning paddle and grinded down to a thickness of 50 µm. Next, the multiprep arm was set to wedging, upon which the sample was further grinded until a wedge was formed. The wedge angle was set to 2°; a higher angle reduces the thin area to see under the microscope and a smaller one would put the integrity of the sample at risk. Wedging was performed with foils of 3 µm grain size until the Si substrate turned orange when transmitting light under an optical microscope. Then, the pad was switched to 1 µm and lubricated with green lube (Struers, Copenhagen, Denmark). This grinding was continued until interference fringes could be distinguished on the sample under a light microscope. The specimen was then detached from the multiprep paddle and glued onto a Cu ring.

The last step was ion polishing with $Ar^+$ ions on a precision ion polishing system (PIPS, Gatan Inc., Pleasenton, CA, USA). The procedure started with 20 min at high energy and high incidence angle (4 keV and ±7°), followed by 10 min at 2 keV and 10 min at 1 keV, respectively, both steps at ±4°. This method has already proven successful for making TEM samples of thin films [25] and bulk ceramics [11]. A schematic of the described sample preparation is provided on the left side of Figure 1.

### 2.3. FIB Lift-Out Sample Preparation

For preparing a sample via FIB, an Auriga SMT SEM (Zeiss AG, Oberkochen, Germany) with a Cobra FIB dual beam system (Orsay Physics, Fuveau, France) was used. The preparation was performed using the lift-out technique [33]. Initially, an area of interest was marked by depositing Pt from a precursor provided from a gas injection system. The Pt was deposited along a 15 µm line and had a thickness of around 100–500 nm. Further, a lamella was cut. First, a trench along one side of the Pt line was milled using $Ga^+$ ions accelerated at 30 keV and a beam current of 20 nA. The ion column was oriented perpendicular with respect to the sample surface. A 20 µm × 15 µm rectangular area was milled along the Pt line. The same process was repeated on the other side of the Pt deposit, with the sample stage rotated by 180°. To perform the undercut and finally free the lamella, the sample was tilted so the $Ar^+$ column forms a 47° angle with the surface of the sample, by doing so the bottom was exposed. At this point, the formed wedge-shaped lamella was still attached to the rest of the substrate by the sides and the bottom. To free the lamella, first the sides were milled. Then an Omniprobe micromanipulator tip (Oxford Instruments, Abingdon, UK) was welded to the lamella by Pt deposition. Finally, the bottom of the lamella was milled, so it could be released safely, its size being around 8 µm in height and 15 µm in width.

The next step was to transfer and fix the lamella to the tip of an Omniprobe grid (Oxford Instruments, Abingdon, UK) and thin down the area of interest. First, the base of the lamella was welded at the end of the microtip with Pt. Once the sample was secured,

the ion column was set parallel along the lamella. To thin the lamella, 30 keV ions at 600 pA were used to remove material on both sides, this step was repeated again decreasing the current down to 50 pA. The final polishing was performed using 5 keV at 100 pA, again on both sides, until the area was bright when imaged using the SEM indicating electron transparency. The thinned area was around 3 μm in height and 7 μm in width. The prepared sample stemmed from an area of the film with an Au electrode on top. The here-described FIB lift-out method is widely used on bulk ceramics [34] and thin films [26], as it is universally applicable and does not require any other dedicated TEM preparation facilities. A flowchart of the sample preparation can be seen on the right side of Figure 1.

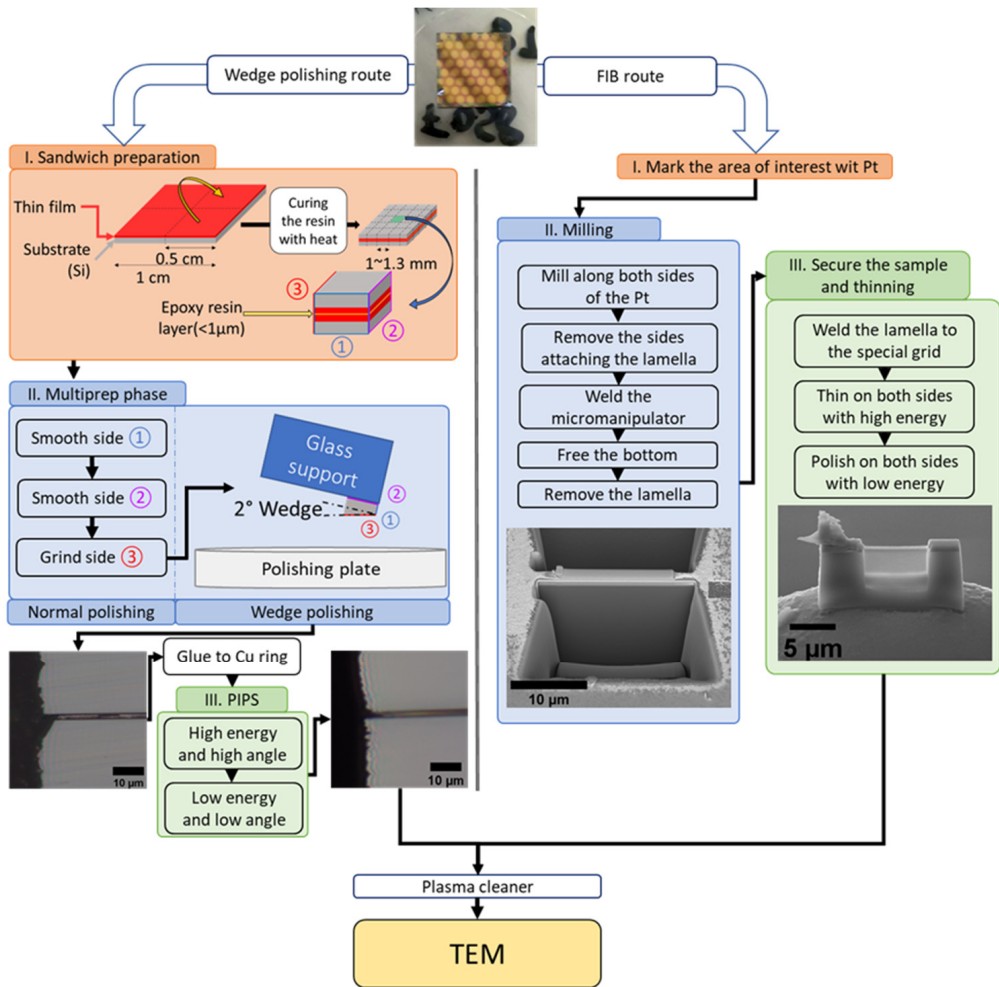

**Figure 1.** Flowchart for TEM sample preparation via wedge polishing (**left side**) and FIB lift-out (**right side**).

*2.4. TEM Observation*

The TEM observations were performed on a Cs corrected JEOL 2100F (JEOL Ltd., Tokyo, Japan) operating at 200 kV. Before introducing any sample into the microscope, the sample holder loaded with the specimen was cleaned in a plasma cleaner Model 1020 (Fischione Inc., Export, PA, USA) for 5 min using a 75% argon and 25% oxygen gas mixture. A double tilt holder (JEOL Ltd., Tokyo, Japan) was used. At the beginning of the analysis ADF STEM (annular dark field scanning transmission electron microscopy) images with a Gatan detector and EELS (Electron Energy Loss Spectra) were taken to measure the thickness of the sample using a Gatan Imaging Filter (GIF) (Gatan Inc., Pleasanton, CA, USA). Bright field micrographs were taken to analyze the overall quality of the sample. High resolution images were acquired along the [110] zone axis to analyze the crystal lattice

and its possible distortions. This work used indexes based on the cubic structure of $BaTiO_3$, as the tetragonal structure is not possible to be distinguished from the cubic one with the methods employed here (the distortion between the two phases is ~1%) [35]. Diffraction patterns were taken to identify secondary phases or amorphous phases that could have formed on the surface during sample preparation. The TEM images were recorded on an Orius SC1000 CCD camera (Gatan Inc., Pleasenton, CA, USA).

### 2.5. Simulations and GPA

HRTEM multislice simulations of the samples were conducted using the software JEMS [36] to check the accuracy of the obtained images. The results were also studied by GPA (geometrical phase analysis) using the software Strain++ [37] to assess if there were lattices distortions evident on either of the samples.

### 3. Results

### 3.1. Thickness of the Samples

Before going into detailed analysis, the thickness of both samples was measured to ensure comparability. The results can be seen in Figure 2. On the sample prepared by wedge polishing, the thickness at the edge of the sample is 20 nm at the top (Figure 2b) and increases at a rate of around 0.5 nm for every nm advanced towards the substrate. In the case of the sample prepared by FIB (Figure 2d), the thickness of the top of the film, at the interface with the electrode, is also 20 nm, and it increases when going towards the Pt layer at a rate of 0.35 nm added thickness per nm advanced along the sample, similar to its wedge-polished counterpart. It must be noted that, on the FIB prepared sample, the area with the thinnest material is always along the Au electrode, while on the wedge-polished sample the thin area extends along the surface cleaned by the PIPS, as can be seen in Figure 2a. This means that thin areas are present along the whole extent of the film, therefore observations of thin material are possible near the Pt substrate or the Au electrode. It must be noted that, in principle, parallel FIB samples can be made with a uniform thickness over the whole specimen, but it can be complicated, especially on materials with multiple layers, as the different yields and sputtering rates of each material increase the complexity of the sample preparation. On the other hand, the PIPS will always leave this thickness gradient that provides thin areas to be observed on all exposed layers. In this work, the HRTEM on the wedge-polished sample was performed on an area where only the Au electrode has been removed by the PIPS, i.e., the top of the film. Another detail that must be remarked upon is the extension of the thin area: on the FIB prepared sample it is about 7 μm, as that is the area thinned on the cut lamella; on the other hand, the wedge-polished sample is only thin near the edge of the sample, and this can be around 1 μm in this case. Considering that there is a thin film on each side of the sandwich, that makes around 2 μm of area that is appropriate for HRTEM observations. In summary, the wedge-polished sample presents more variability of thin areas within the multilayer structure of the sample, but the FIB prepared sample presents a more extensive thin area.

### 3.2. Wedge Polished Sample

The TEM analysis on the $BaTiO_3$ thin film prepared by wedge polishing is presented in Figure 3. As can be seen on the bright field (BF) micrograph (Figure 3a), the thin film and the different layers of the substrate are perfectly visible. The film itself is composed of an array of columnar grains of 30~80 nm width and a height of ≈140 nm. The gold electrode is visible on the top left side of the picture, whereas on the top right side it was removed by the PIPS step. When focusing on an individual grain, a layered structure can clearly be seen (Figure 3b), especially in the area near the electrode at the left, where the sample is slightly thinner. These layers are around 7 nm thick. When dividing the thickness by this value the result is that there are approximately 18 layers, which corresponds well to the layers deposited during the CSD process. On the area near the Pt substrate at the left we can see small grains that overlap with the main columnar grain and thus complicate

the analysis. These small grains were likely formed during the early stages of film growth and are overgrown by the columnar grains at later stages. These overlaps complicate the analysis of the sample near the substrate. Such observations were reported already on films prepared in a similar fashion [35]. Figure 3c shows a selected area diffraction pattern (SADP) that was recorded to check the presence of secondary phases in the material. Notably, all the spots can be tied to polycrystalline $BaTiO_3$ with a pseudo-cubic structure, which means that no unwanted phases or compounds were formed during the preparation process. Using high resolution TEM (Figure 3d) the atomic columns can be seen, in this case for a grain oriented along the [110] zone axis. Two twin variants are identified in this micrograph (labelled I and II), separated each by a layer of hexagonal phase. The formation of this phase is due to a combination of the high annealing temperature and the strain exerted by the substrate on the film during processing. These phenomena were reported for this material in previous works [38,39]. The image allows us to distinguish the different A site atomic columns (Ba), as well as structural details of the hexagonal phase, for example how in every orientation of the zig-zag pattern there are four A-site atomic columns. The perovskite and hexagonal structures are more evident when a crystal model is overlaid on the image. A Fast Fourier Transform (FFT) of the area containing the hexagonal phase is shown in Figure 3e; as can be seen, it contains reflections from the two variants. Such reflections correspond to a $[011]_{pc}$ zone axis; the spots are slightly diffuse, indicating a slight degree of disorder, probably caused by the $Ar^+$ bombardment during the PIPS phase. Generally, it should be noted that the quality of the images and the clearly distinguishable nanostructured features prove that the wedge polishing technique followed by ion milling is a viable technique for TEM sample preparation of polycrystalline thin films. The quality of the sample allows us to even take HRTEM images where multiple phases can be seen.

### 3.3. FIB Prepared Sample

The TEM analysis of the thin film sample prepared by FIB is shown in Figure 4. The results look very similar to the sample prepared by wedge polishing. Again, the different layers of the sample and the polycrystalline, columnar microstructure of the film are both visible (Figure 4a). Above the Au electrode, a remainder of the deposited Pt used during the lift-out preparation can be seen. When focusing on a grain, the sol-gel processed layers (Figure 4b) can also be identified as previously in Figure 3b. Furthermore, the diffraction pattern shown in Figure 4c is in accordance with the previously shown case: the different spots can be tied to the pseudo-cubic $BaTiO_3$ structure. HRTEM imaging on a [110] oriented grain was also possible (Figure 4d). Two twin [110] variants (marked I and II) separated by a hexagonal phase were again identified and the A-site atoms were discernible. In this case, the FFT of the hexagonal phase was also calculated and is shown on Figure 4e, showing that it has [011] reflections of both variants. In this case, slightly diffused spots are also visible, this can be an effect of ion bombardment where a slight disorder is created in the near surface lattice, as was also the case in Figure 3e. Overall, the sample quality of the FIB prepared specimen is suitable for identifying its grain structure and perform atomic resolution imaging. These results prove that FIB lift-out is also a viable method to prepare TEM samples of thin films where atomic resolution is achievable.

### 3.4. Simulations and Strain Analysis through GPA

The HRTEM results on both samples were compared with multislice simulations conducted using the JEMS software, and the results (Figure 5) matched the thickness measured on EELS, although the conditions of focusing were different. GPA analysis was also performed on both materials to check if the preparation could modify the local strain of the film (Figure 6); however, on both samples no remarkable distortion differences from one sample to another were evidenced.

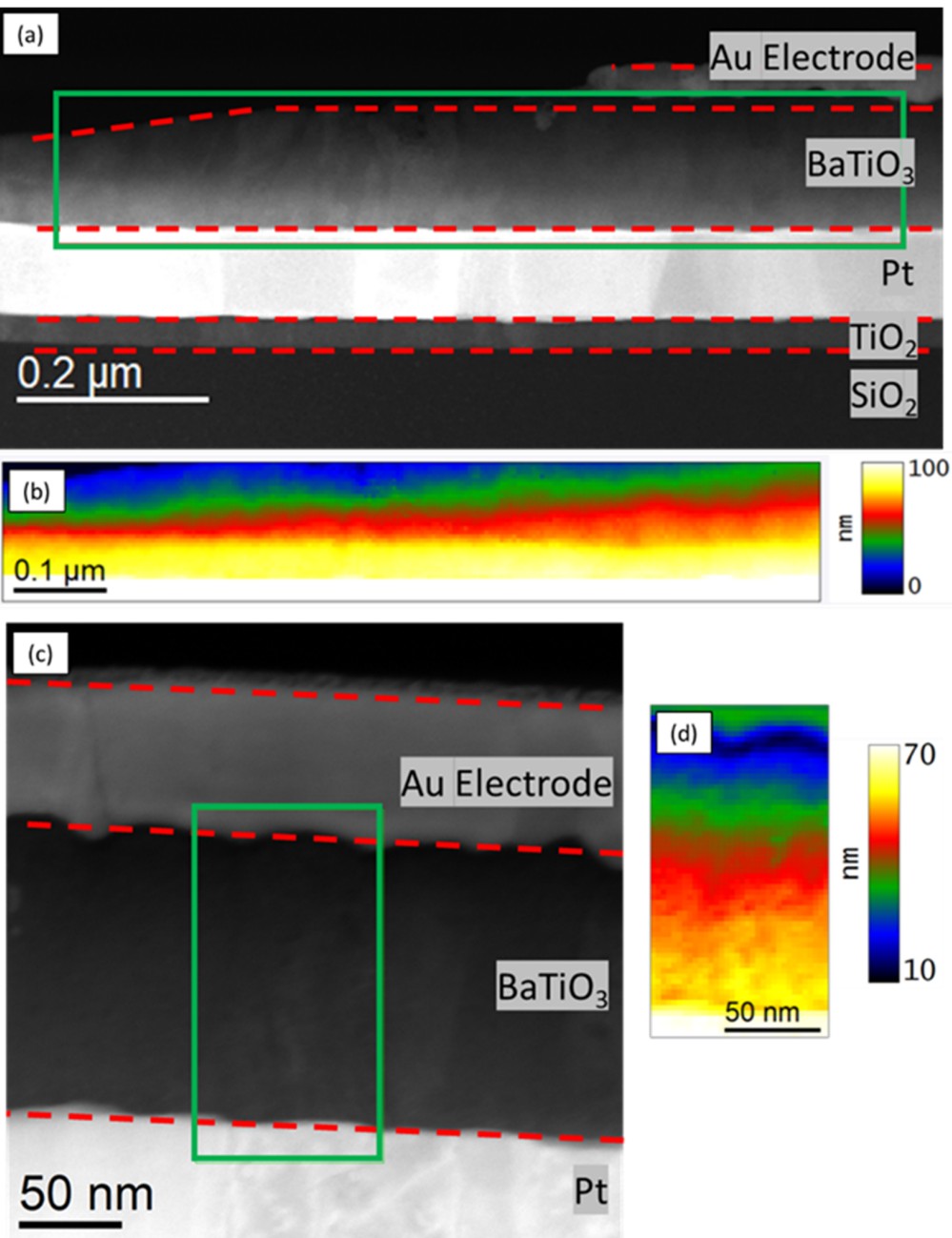

**Figure 2.** (**a**) STEM HAADF (high angle annular dark field) image of the BaTiO$_3$ sample made by wedge polishing. (**b**) Thickness map measured by EELS within the green rectangle on (**a**). (**c**) Image of the sample made by FIB. (**d**) Thickness map measured by EELS within the green rectangle on (**c**).

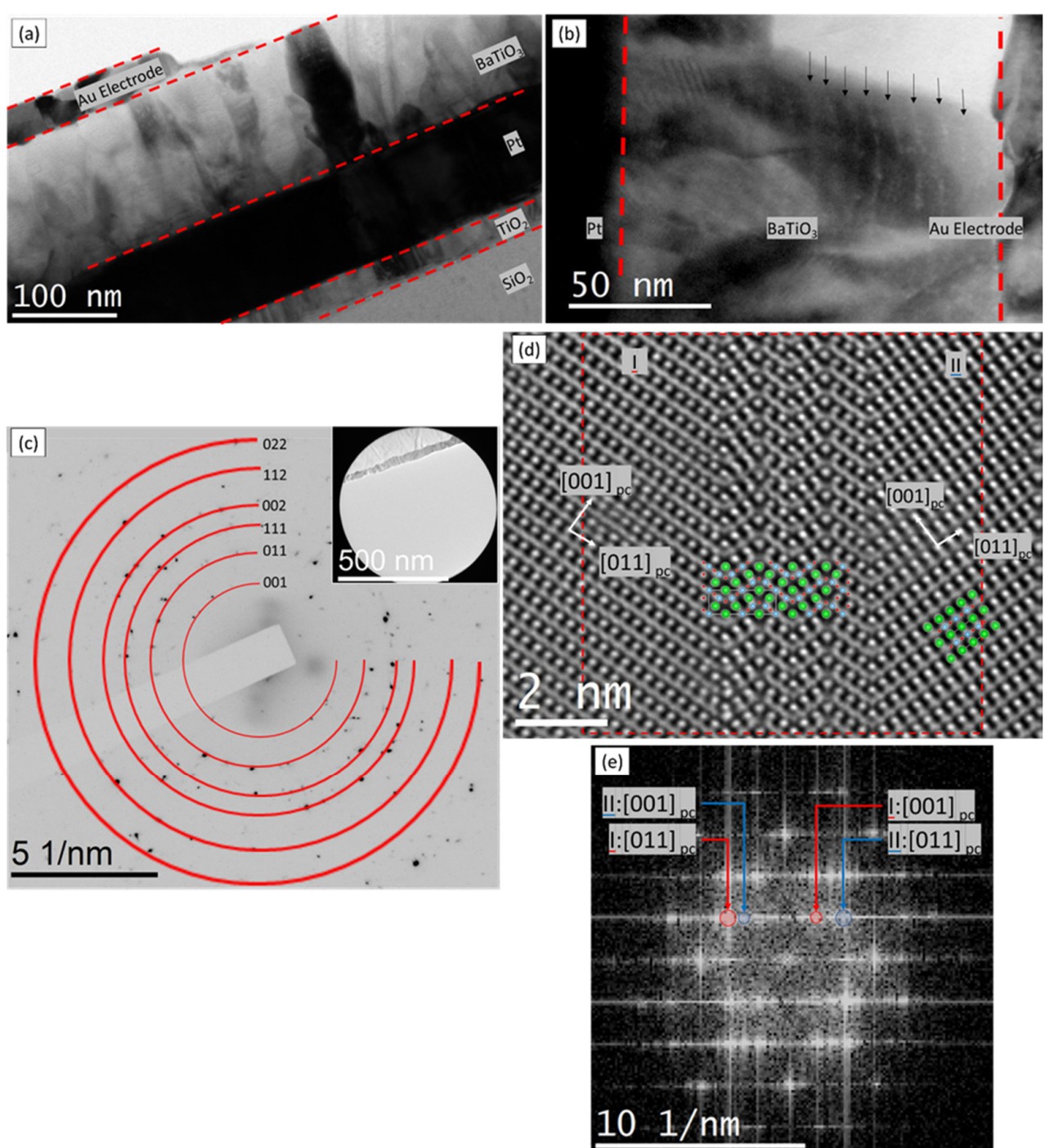

**Figure 3.** TEM analysis on a BaTiO$_3$ thin film sample prepared by wedge polishing followed by PIPS. (**a**) Brightfield (BF) image depicting the general film layout, showing also the different substrate layers. (**b**) BF image of an individual grain in the film, indicating that they are columnar; the deposition layers are also evident, their separation marked with black arrows. (**c**) SADP of the film, with red circles marking the expected positions of diffraction peaks for the BaTiO$_3$ cubic structure. The inset shows the area selected used to take the pattern. (**d**) HRTEM of a grain with a [110] zone axis orientation, presenting two variants separated by a hexagonal phase. Two illustrations of the hexagonal and cubic BaTiO$_3$ patterns are overlaid to confirm the structure. (**e**) FFT of the red square shown on (**d**).

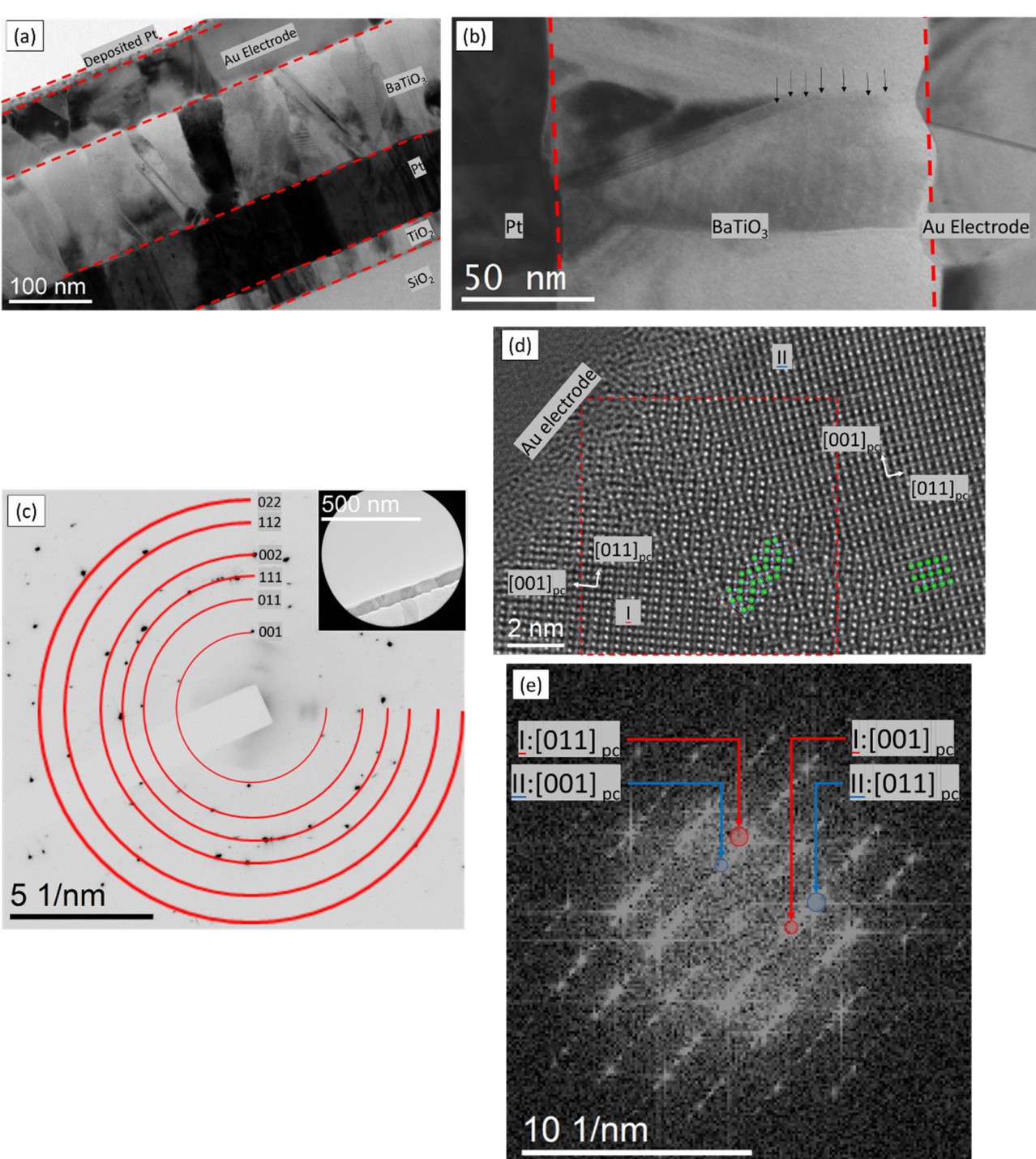

**Figure 4.** TEM analysis on a BaTiO$_3$ thin film sample made by FIB lift-out. (**a**) BF image of the general film structure, showcasing also the different substrate layers. (**b**) BF image of a grain of the film, demonstrating that they are columnar; the deposition layers are also evident, their separations are marked with black arrows. (**c**) SADP of the film, the red circles mark the positions matching the diffraction peaks of the BaTiO$_3$ cubic structure. The inset shows the area selected for diffraction. (**d**) HRTEM of a grain with a [110] zone axis orientation, we can see two variants separated by a hexagonal phase. Overlaids of the hexagonal and cubic BaTiO$_3$ patterns are displayed to confirm the structure. (**e**) FFT of the red square represented on (**d**).

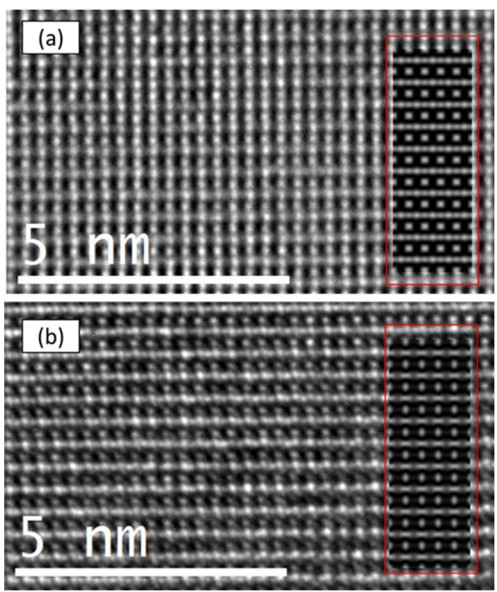

**Figure 5.** HRTEM images of the samples made by (**a**) wedge polishing and (**b**) FIB along the [110] zone axis. The red rectangles contain an overlap of the result of an image simulation on BaTiO$_3$; the thickness of the sample is 24 nm and the defocusing was set to 40 nm and 50 nm, respectively.

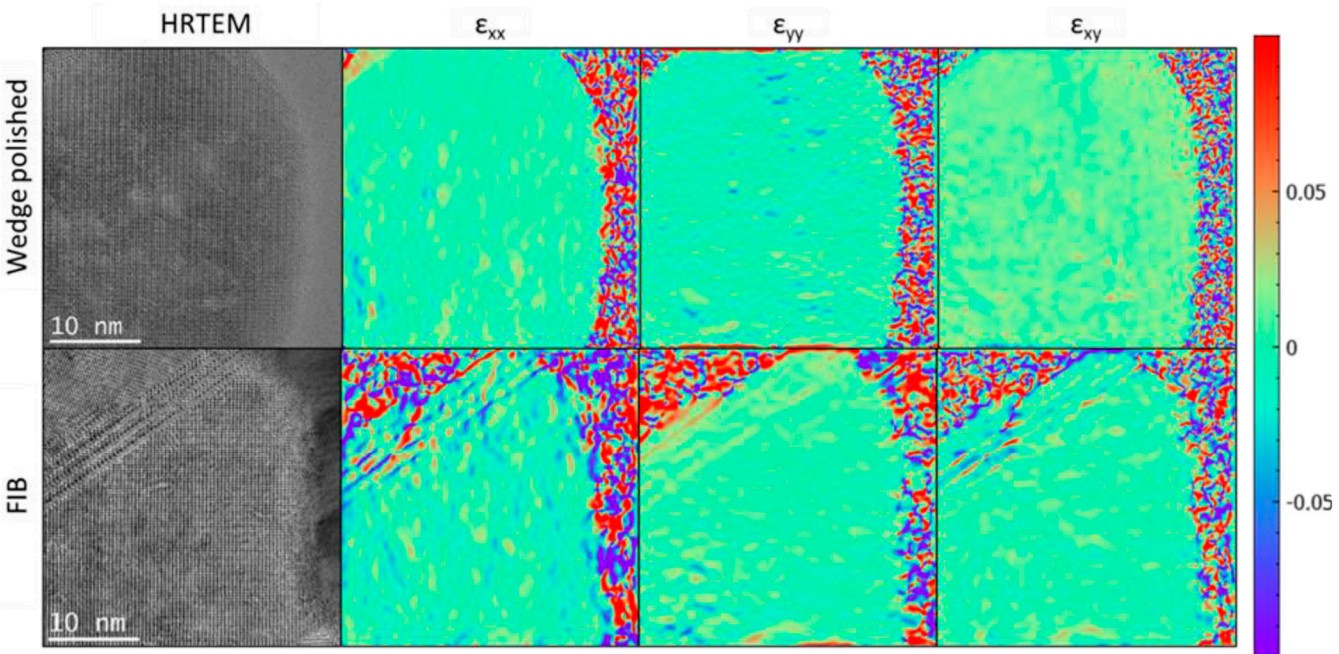

**Figure 6.** GPA strain profiles calculated from both samples on [110] zone axis orientation. The *x* axis corresponds to the [100] direction and the *y* axis to the [110] direction.

## 4. Discussion

The compared methods for TEM sample preparation, wedge polishing and FIB lift-out, have demonstrated the capability to produce samples of very similar quality with a similar thickness. In both cases, high resolution atomic imaging is achievable and it is possible to see phase changes within an area of a few square nanometers with perfect detail. Considering this, it is fair to say that both methods are equally viable for the sample preparation of ceramic thin films made by CSD. Nonetheless, the wedge polished sample has the advantage of having a thickness gradient oblique to the substrate's surface. This allows for thin regions to be available over the whole thickness of the film, whereas on

the FIB prepared sample the thin area is only limited to the surface, albeit having a higher extension. Given these small differences in quality, further considerations may play a role when deciding which method to use for TEM sample preparation in the case of ceramic thin films: in particular, cost and availability of the method. The equipment and consumables necessary for wedge polishing require very little investment: a multiprep machine or a manual tripod polisher with a standard polishing wheel [32,40], diamond foils, green lube, and an ion polishing system. The consumables can be used for multiple samples: if taken care of, a set of foils can be used indefinitely. Timewise, the initial polishing phase typically has a duration of about two hours, whereas the PIPS step lasts for less than one hour. FIB lift-out, on the other hand, is more expensive, as it relies on special equipment (a combined SEM-FIB). Considering the running/depreciation costs of such a machine and the need for a skilled operator, the cost of preparation per sample can be considerably higher (a factor 10 at least) compared to wedge polishing. In terms of time, however, an experienced FIB operator can make a sample in about two hours, a similar time scale as required for the wedge polishing. Both methods have a high repeatability, especially when performed by experienced professionals with a consistent methodology. Certainly, it can be easier to come to a reliable methodology on a computer controller FIB system, where consolidation of the process might become faster. On the other hand, for wedge polishing it might be harder to achieve the appropriate conditions to reproduce the sample preparation reliably, as the operator needs to become familiar with the machinery. For FIB, in addition, another practical aspect must be considered; an SEM/FIB system is not always available in all research centers; most groups sometimes rely on third parties with FIB capabilities. Wedge polishing equipment, on the other hand, is readily available in virtually all research centers. Since this type of preparation can be done on a conventional polishing wheel with a manual tripod polisher, there is no specific need for a multiprep. Such polishing tools can be obtained literally anywhere, for instance by a 3D printer just downloading related schematics and printing the components [41]. This could make this method more popular for researchers looking for a way to prepare high quality TEM samples at a low cost. Although the cost of a PIPS should not be overseen, these units are generally more available and easier to use and maintain than an SEM/FIB system.

## 5. Conclusions

TEM samples of BaTiO$_3$ thin films were prepared by two methods—wedge polishing followed by PIPS and FIB lift-out, respectively. The samples were analyzed by TEM using brightfield (BF) imaging, diffraction measurements, atomic resolution microscopy and thickness measurements via EELS, evidencing that both methods are able to prepare samples with the same quality, reaching down to atomic resolution capability, although on the FIB prepared sample the thin area is limited to the surface of the film. These results make both techniques equally valid for TEM sample preparation of thin films, but their availability might differ considerably. FIB requires expensive equipment that must be operated by experienced personnel. On the other hand, wedge polishing is a more affordable and readily available technique.

The results of this work might help groups developing thin films to newly approach TEM based characterization, which is nowadays increasingly relevant due to the development of miniaturized devices.

**Author Contributions:** Conceptualization, D.K.; methodology, J.S.-M.; validation, J.S.-M.; formal analysis J.S.-M.; investigation, J.S.-M.; resources, B.S. and F.B.; data curation, J.S.-M.; writing—original draft preparation, J.S.-M.; writing—review and editing, D.K. and M.D.; visualization, J.S.-M.; supervision, D.K. and M.D.; funding acquisition, M.D. All authors have read and agreed to the published version of the manuscript.

**Funding:** Funding from the European Research Council (ERC) under the European Union's Horizon 2020 research and innovation program (grant agreement No. 817190) is acknowledged.

**Data Availability Statement:** The data presented in this study are available on request from the corresponding author.

**Conflicts of Interest:** The authors declare no conflict of interest.

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
