# Peer review of "FIB and Wedge Polishing Sample Preparation for TEM Analysis of Sol-Gel Derived Perovskite Thin Films"

_ceramics, doi:10.3390/ceramics5030023_

Round 1

Reviewer 1 Report

This is an expertly written work which is very useful for experimentalists. It may be published after minor revision.

Two points need to be clarified:

1) p. 3 row 91: A native SiO2 layer is usually very thin about 0.5 nm. A 500 nm thick SiO2 film was probably manufactured by thermal oxidation. Here, the authors have to specify was it wet oxidation (with regard to the thickness more probable) or dry oxidation.

2) The authors have mentioned the impact of ion bombardment during ion milling in the introduction. However, this aspect is not discussed in section 3, although the fast fourier transforms in Figures 3e and 4e indicate a certain disorder (at least not an ideal crystal lattice).

Author Response

Dear reviewer,

Thanks for your helpful comments, concerning the points that need to be clarified:

1) The substrates are commercially available, the company that produces them (SINTEF) did not give us specific details on how they were made, mostly because they want to keep their methods. I wrote on the draft that they are commercially available and the company that produces them.

I attach the corrected version considering your comments and the ones from the other reviewer.

Thanks for your help,

Best regards,

Jorge

Reviewer 2 Report

The manuscript entitled “FIB and wedge polishing sample preparation for TEM analysis of sol-gel derived perovskite thin films” presents the comparison of TEM results obtained on wedge polished and FIB treated samples of a thin polycrystalline barium titanate film prepared by chemical solution deposition. The comparison of both the method and their clear description can be considered as the main contribution of the manuscript.  It is shown that both the method provide us with samples  making possible to carried out high resolution atomic imaging and to see phase changes within an area of few square nanometers. However, it is also shown that the FIB technique can be discriminate by its limited availability at laboratories and due to economical reasons.  Thus, the manuscript deals with the affordability of both the methods and not with comparing their repeatability which should be also discussed. In addition, only minor changes have to be made in the manuscript, namely:

11.       On Page 3 top , a reader can find a sentence “After mixing them, the volume was adjusted to obtain a solution with a concentration of 0,3 M, which was stored in the fridge until ready to be used” which has to be improved. From this sentence it is not clear if cooling in a fridge (please specify a type and temperature) is necessary step for the sol preparation or if the sol has to be kept in the fridge in order to avoid its deterioration. Did you apply the cooled sol or sol having the laboratory temperature?

22.       On Page 7 top, it written “ On the area near the Pt substrate at the right …” which is not true because the Pt substrate is on the left side of the figure.

33.       The resolutions of discussed details in Figs. 3b and 4b are not very well. Could you improve them?

Author Response

Dear Reviewer,

Thanks for your helpful comments, please find below our responses concerning the points that need to be clarified:
The repeatability of these methods is consistent in both cases, especially when
performed by experienced professionals on the field, which has been commented on in
section 4.

1) The solution can be stored in the fridge to extend its lifetime. It is not really easy to say for how long exactly, as our team works with different solutions that last differently. In the case of the BaTiO3, it can range to weeks or even months. This sentence on section 2.1 has been corrected to be more specific.

2) This has been corrected, “right” has been changed with “left” (section 3.2)

3) The contrast of the layers on the Figs. 3b and 4b is a compromise between seeing
the maximum of layers while at the same time not oversaturating the image. We are aware that they look really dim, hence the arrows to help the reader notice them. Please note that they also appear weak in the work of J. L. Cia [35]. Concerning the overlap of the grains near the substrate of Fig. 3b, it is hard to see them due to the grain boundaries not being edge on. But the point is that having multiple grains overlying complicates the analysis of the sample near the substrate. This consideration has been added to section 3.2. Should characterization of the thin film-substrate
interface be needed, then the observations would be done closer to the edge where that area is thinner and can provide better results. This is already mentioned in section 3.1 for the wedge polished sample.

I submit the corrected version considering your comments and the ones from the other reviewer.

Thanks for your help,
Yours sincerely,
Jorge Sanz Mateo (on behalf of all authors)
